# The impact of changing cigarette smoking habits and smoke-free legislation on orofacial cleft incidence in the United Kingdom: Evidence from two time-series studies

**Matthew Fell**[1]*, **Craig Russell**[2], **Jibby Medina**[3], **Toby Gillgrass**[2], **Shaheel Chummun**[4], **Alistair R. M. Cobb**[4], **Jonathan Sandy**[1], **Yvonne Wren**[1], **Andrew Wills**[5©], **Sarah J. Lewis**[6©]

1 Cleft Collective, Bristol Dental School, University of Bristol, Bristol, United Kingdom, 2 Scottish Cleft Service, Royal Hospital for Children, Glasgow, United Kingdom, 3 Clinical Effectiveness Unit, Royal College of Surgeons of England, London, United Kingdom, 4 South West Cleft Service, University Hospitals Bristol and Weston NHS Trust, Bristol, United Kingdom, 5 Faculty of Health Sciences, University of Bristol, Bristol, United Kingdom, 6 Medical Research Council Integrative Epidemiology Unit, University of Bristol, Bristol, United Kingdom

© These authors contributed equally to this work.
* mattfell@doctors.org.uk

## Abstract

### Background

Both active and passive cigarette smoking have previously been associated with orofacial cleft aetiology. We aimed to analyse the impact of declining active smoking prevalence and the implementation of smoke-free legislation on the incidence of children born with a cleft lip and/or palate within the United Kingdom.

### Methods and findings

We conducted regression analysis using national administrative data in the United Kingdom between 2000–2018. The main outcome measure was orofacial cleft incidence, reported annually for England, Wales and Northern Ireland and separately for Scotland. First, we conducted an ecological study with longitudinal time-series analysis using smoking prevalence data for females over 16 years of age. Second, we used a natural experiment design with interrupted time-series analysis to assess the impact of smoke-free legislation. Over the study period, the annual incidence of orofacial cleft per 10,000 live births ranged from 14.2–16.2 in England, Wales and Northern Ireland and 13.4–18.8 in Scotland. The proportion of active smokers amongst females in the United Kingdom declined by 37% during the study period. Adjusted regression analysis did not show a correlation between the proportion of active smokers and orofacial cleft incidence in either dataset, although we were unable to exclude a modest effect of the magnitude seen in individual-level observational studies. The data in England, Wales and Northern Ireland suggested an 8% reduction in orofacial cleft incidence (RR 0.92, 95%CI 0.85 to 0.99; P = 0.024) following the implementation of smoke-free legislation. In Scotland, there was weak evidence for an increase in

**Data Availability Statement:** All relevant data are within the paper and its Supporting Information files.

**Funding:** All authors have completed the ICMJE uniform disclosure form at www.icmje.org/coi_disclosure.pdf and declare: no support from any organisation for the submitted work; MF is supported by the VTCT Foundation for a research fellowship with the Cleft Collective at the University of Bristol; SL is supported by a project grant from the Medical Research Council (MR/T002093/1); no financial relationships with any organisations that might have an interest in the submitted work in the previous three years; no other relationships or activities that could appear to have influenced the work. Funders had no role in the study design; in the collection, analysis and interpretation of data; in the writing of the report; and in the decision to submit the article for publication. The authors can confirm their independence from funders and that all authors had full access to all of the data (including statistical reports and tables) in the study and can take responsibility for the integrity of the data and accuracy of the data analysis.

**Competing interests:** The authors have declared that no competing interests exist.

orofacial cleft incidence following smoke-free legislation (RR 1.16, 95%CI 0.94 to 1.44; P = 0.173).

## Conclusions

These two ecological studies offer a novel insight into the influence of smoking in orofacial cleft aetiology, adding to the evidence base from individual-level studies. Our results suggest that smoke-free legislation may have reduced orofacial cleft incidence in England, Wales and Northern Ireland.

## Introduction

Orofacial clefts (OFC) of the lip and/or palate are common congenital anomalies with a complex aetiology, based on interactions between environmental and genetic factors [1,2]. Cigarette smoking is a common environmental factor with evidence of a causal role in OFC aetiology [3]. Meta-analyses of studies assessing the relationship between active maternal smoking during pregnancy and OFC indicate a moderate effect (OR 1.42, 95% CI 1.27 to 1.59) [4]. A similarly positive association is seen with meta-analyses of studies investigating the relationship between passive maternal smoking (also known as second-hand smoking or environmental smoking) and OFC (OR 1.54, 95%CI 1.11 to 2.12) [5,6].

Conventional observational studies to establish a causal relationship between smoking and OFC have inherent limitations such as selection and recall bias, and confounding [7]. Randomised controlled trials are neither feasible nor ethical [8]. Ecological studies, utilising data from national registries, offer an alternative approach, with different sources of bias, and can triangulate evidence by testing hypotheses in a real-world setting [9–11]. Given the associations reported from individual level studies, it is relevant to explore these at the population level.

The UK, with a large population, is well placed for a population-level evaluation of OFC incidence. Cleft services were centralised within the National Health Service in 1998 and there was a subsequent creation of national cleft registries in 2000 [12–14]. Smoking habits have changed in the UK population over the last two decades, with a decline in the proportion of people actively smoking in part through the implementation of a series of anti-smoking government policies [15]. The implementation of comprehensive smoke-free legislation across the UK between 2006–2007 is likely to have reduced exposure to passive smoking [16], with a 27% mean reduction in cotinine levels amongst non-smoking adults [17] and an improvement in air quality in public bars [18]. The aim of this study was to use two time-series designs, namely an ecological study and a natural experiment, to determine whether there is any evidence that changing exposure to smoking in the UK has had an impact on OFC incidence.

## Methods

### Study design and setting

Two time-series studies were performed using routinely collected data from the UK between 2000 to 2018. First, an ecological study with longitudinal time-series analysis to investigate the association between the proportion of pregnant women actively smoking and OFC incidence. Second, a natural experiment design with interrupted time-series analysis was used to assess the impact of the implementation of smoke-free legislation on OFC incidence. We followed

published guidance relating to ecological studies [19] and natural experiments [9,20] and reported this study using the STROBE framework, where possible to do so [21].

## Outcome variables

The main outcome of interest was annual incidence of OFC, defined as the total number of children born with any subtype of cleft lip and/or palate in the UK per 10,000 live births each year. Incidence data were also obtained for the subtypes cleft lip with or without palate (CLP) and cleft palate only (CPO), to explore whether these subgroups had different associations with smoking, as others have found [22,23].

The annual calendar year count of live births with OFC in England, Wales and Northern Ireland (EWNI) was accessed from the Cleft Registry and Audit NEtwork (CRANE) Database from 2000 to 2018 [13]. Case ascertainment in the CRANE Database for OFC registration in EWNI is 95% [24]. Annual counts of all live births with OFC in Scotland was obtained from the clinical database of the National Cleft Service for Scotland, which reported by financial year (April to April) between 2000–2009 and calendar year between 2010–2018. Children born in Scotland between January-April 2010 were counted twice due to the change in annual reporting intervals, but exclusion of 2010 data in the analysis did not impact the results. The difference in time interval reporting structures between the two UK cleft datasets precluded data amalgamation, thus outcome data from Scotland was considered separately. OFC sub-types are identified nationally according to the LAHSHAL classification [25].

OFC incidence in EWNI was calculated using the number of children born with OFC and registered on CRANE, divided by the total number of live births, reported by the Office of National Statistics (ONS) for England and Wales [26] and the Northern Ireland Statistics Research Agency [27]. OFC incidence in Scotland was calculated in the same way using the Scottish clinical database and live birth data reported by financial year in the Scottish Morbidity Record between 2000–2009 [28] and reported by calendar year by Scottish National Records between 2010–2018 [29].

## Ecological study exposure variables and co-variables

The exposure of interest in the ecological study was active maternal cigarette smoking during the first trimester of pregnancy, because this coincides with the embryological formation of facial structures and is thus regarded as the critical time for smoking to play a role in OFC aetiology [30–32]. We could not identify a nation-wide data source to estimate UK prevalence of maternal smoking in the first trimester [33]; therefore, two aggregate proxy measures were used. The primary exposure proxy measure was the proportion of active cigarette smokers in females over 16 years in the UK (Exposure Proxy 1) by calendar year, reported by the ONS Annual Population Survey, which has an annual sample size of 320,000 and aims to be broadly representative of the UK population [34,35]. As a sensitivity analysis, the proportion of active smokers amongst all pregnant women attending an antenatal booking appointment in Scotland by financial year was used as a secondary proxy measure (Exposure Proxy 2), reported by Public Health Scotland in the Scottish Morbidity Record [36].

Aggregate time-series data on potential confounders over the 2000–2018 period of the study was sought, including maternal age, maternal alcohol consumption, maternal body mass index and folic acid supplementation [37–40]. Maternal age was the most reliably reported co-variable at a population level and was available for populations in both EWNI [26,27] and Scotland [29] throughout the study period. Other co-variables were either not accurately measured or not available, so were not used in the adjusted model.

## Natural experiment intervention and bias

The single intervention under investigation in the natural experiment analysis was the implementation of smoke-free legislation in the UK, which prohibited smoking in the workplace and enclosed public spaces [41]. We tested the impact of the 2007 legislation in EWNI (implemented in Wales 2nd April 2007, Northern Ireland 30th April 2007 and England 1st July 2007) and legislation in Scotland implemented 26th March 2006 against 19 annual cleft incidence data points between 2000–2018. The risk of bias in the natural experiment was assessed according to the Cochrane Effective Practice and Organisation of Care (EPOC) criteria [42].

We anticipated the impact model of the smoke-free legislation a priori to involve a level change reduction in OFC incidence, due to the well-defined dates of the smoking ban implementation, which should have an immediate impact on passive smoking exposure [16]. We did not hypothesise a change in slope of OFC incidence following the legislation, due to the consistent secular trend of decreasing active smoking at the population level present before and after the ban [34]. We assumed a lag period, hypothesising that there is an impact of smoking on OFC aetiology in early pregnancy and accounting for the nine-month human gestation period. In EWNI the 2007 smoke-free legislation was introduced between April-July; all children born at term in 2007 would have been conceived prior to the smoking ban; children born in 2008 may have been conceived before or after the implementation of the ban, therefore 2008 data was excluded from the analysis [43]; children born in 2009 would all have been conceived after the legislation. For Scotland, the smoke-free legislation was implemented in March 2006, and as cleft cases were counted from April to April, the financial year starting in 2006 was involved in the lag phase and was excluded from the analysis. The first financial year in Scotland where all births would have been conceived after the legislation was 2007.

## Statistical methods

All variables were described initially through visualisation and summary statistics with inter-quartile range (IQR). For all analyses, the count outcome was modelled as a quasi-Poisson distribution to account for overdispersion and was offset by total live births to model the time-trend of incidence. Autocorrelation was investigated with plots of residuals [8,44]. The trends in annual OFC incidence were described using cubic flexible spline functions of time with 4 degrees of freedom specified for both EWNI and Scottish datasets to minimise both dispersion and autocorrelation [45]. Effect estimates were reported as incidence rate ratios (RR) with 95% confidence intervals (CI) and p-values. P values were interpreted as continuous measures of the strength of evidence against the null hypothesis.

To estimate the potential impact of active smoking prevalence over time on OFC incidence since the year 2000, we used information from our meta-analysis of individual-level studies, in which we reported a pooled odds ratio between active maternal cigarette smoking and OFC of 1.42 [4], to create predicted OFC incidence projections at a population level. The projection was created using the proportional attributable fraction (PAF), calculated using both Proxy Measure 1 and 2 of active smoking prevalence [46]. A worked example of the PAF projections can be found in S1 Appendix.

The ecological time-series analytical strategy followed the principles described by Bhaskaran et al. (2013), to investigate whether short-term variation in OFC incidence could be explained by changes in the active smoking exposure [47]. To analyse the association between annual OFC incidence and smoking prevalence (for Exposure Proxy 1 and 2 separately) an unadjusted (crude) model was first fitted. The final model was adjusted for the yearly long-term incidence trend and maternal age with a 1-year lag, as an upward approximation to account for the 9-month human gestation period since the model works with annual rates [45].

The natural experiment analytical strategy followed the principles described by Lopez-Bernal et al. (2017), with an interrupted time-series segmented regression model used to compare OFC incidence before and after the implementation of the smoke-free legislation [48]. As justified above, the models were parameterised to allow and test for a change in level of OFC incidence after the legislation. Sensitivity analyses were performed to model a change in level at other non-intervention years (where at least 3 datapoints fall on each side) to assess whether results are just reflecting a better statistical fit to random sampling variation rather than a true finding generated by the introduction of the smoke-free legislation [38].

Raw data are available in S2 Appendix. All analyses were conducted using the R Project for Statistical Computing, version 4.0.5 (http://www.R-project.org/) with code available in S3 Appendix.

## Results

### Secular trends

Of 13,225,737 live births in EWNI between 2000–2018, 20,049 children were born with OFC with an annual incidence per 10,000 live births ranging from 14.2–16.2 (IQR 0.9). Of 1,038,843 live births in Scotland, 1,720 children were born with OFC, with an incidence per 10,000 live births ranging from 13.4–18.8 (IQR 2.4). Visual inspection of OFC incidence revealed annual fluctuation and regression analysis with adjustment for long-term trends (see Fig 1) did not suggest a change over time in either EWNI ($\beta$0.002, SE 0.003; P = 0.616) or Scotland ($\beta$0.000, SE 0.006; P = 0.957). The smaller population in Scotland showed greater variation in the OFC incidence data compared to EWNI.

In EWNI, 10,282 children (51%) were born with CLP and 8, 634 (43%) children were born with CPO subtypes. Unspecified subtypes were recorded for 1,133 children (6%). CLP incidence ranged from 6.7–8.8 (IQR 0.6) and CPO incidence ranged from 5.4–7.4 (IQR 0.7) as seen in Fig 2. In Scotland 886 children (52%) were born with CLP and 834 children (48%) were born with CPO. The incidence of CLP in Scotland ranged from 6.2–10.1 (IQR 1.3) and the incidence of CPO ranged from 5.6–10.0 (IQR 2.3).

The proportion of active smokers according to Exposure Proxy 1 (females over 16 years in the UK) decreased from 25.5% in 2000 to 16.2% in 2018 (IQR 5.9%) and in Exposure Proxy 2 (pregnant females in Scotland) from 28.9% in 2000 to 15.1% in 2018 (IQR 6.9%) as seen in Fig 3. This was a relative reduction of 36.5% for Exposure Proxy 1 and 47.8% for Exposure Proxy 2 during the study period. Linear regression shows strong evidence for a reducing trend in the proportion of active smoking for both Exposure Proxy 1($\beta$-0.61, SE 0.03; P<0.001) and Exposure Proxy 2($\beta$-0.81, SE 0.03; P<0.001). The projected change in OFC incidence in both EWNI and Scotland over the study period, calculated from the PAF predictions, was a relative incidence reduction of 2.8% under Exposure Proxy 1 and 4.1% under Exposure Proxy 2 (See S1 Appendix).

### Longitudinal time series analysis

In both EWNI and Scotland there was no strong evidence of a correlation between OFC incidence and either proxies of active smoking prevalence in the crude or adjusted analysis (See Table 1).

### Interrupted time series analysis

The pre-legislation trends of OFC incidence were positive and linear in both datasets. In EWNI, there was moderate evidence for an 8% relative level decrease in OFC incidence

## England, Wales and Northern Ireland

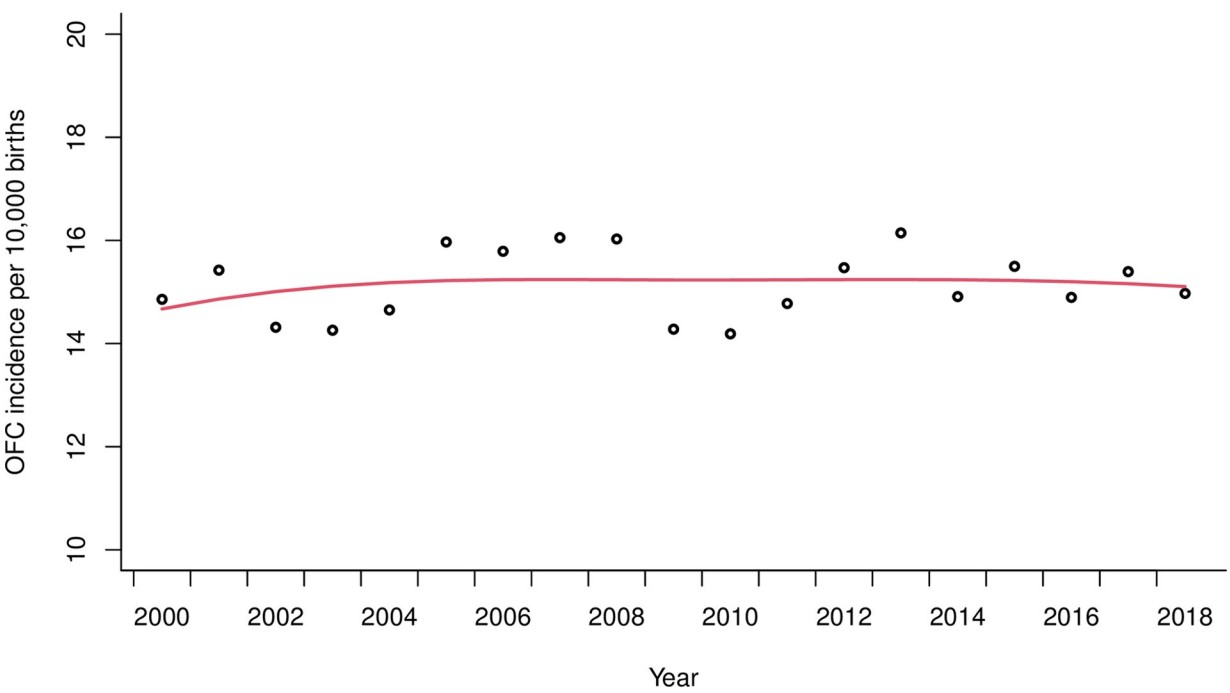

## Scotland

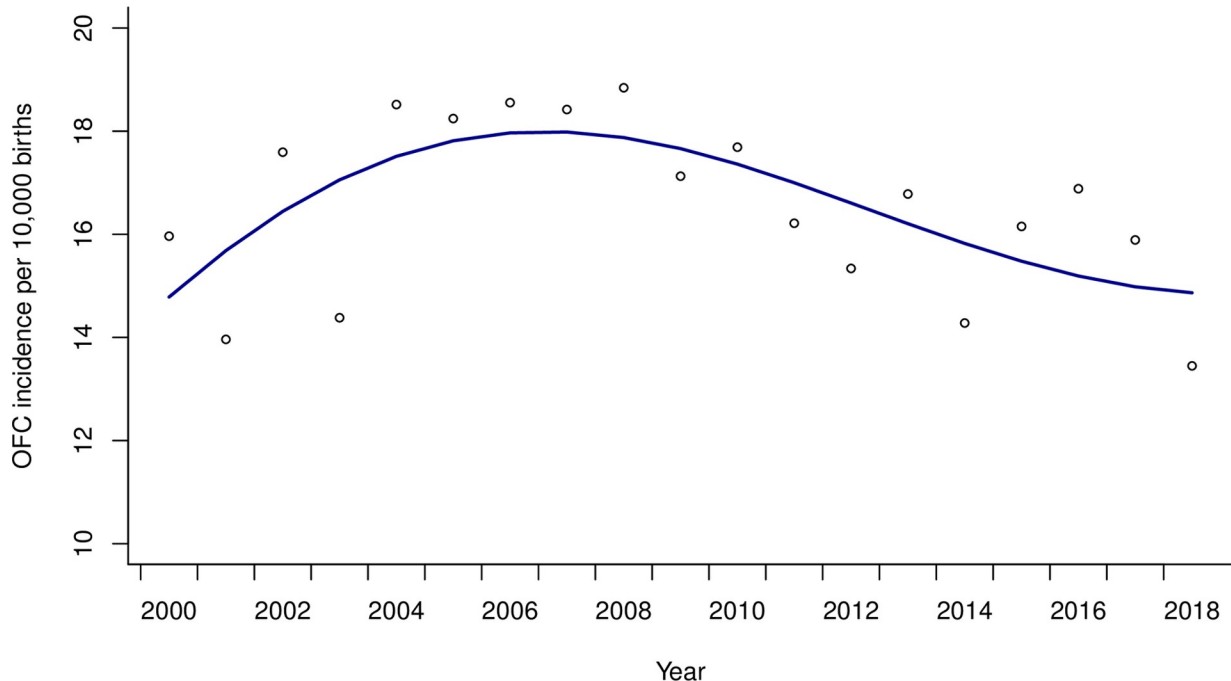

**Fig 1. Quasi-Poisson regression line of orofacial cleft (OFC) incidence over time in EWNI and Scotland with long-term trends modelled using cubic flexible spline functions.** The scatter are the raw annual data points.

## England, Wales and Northern Ireland

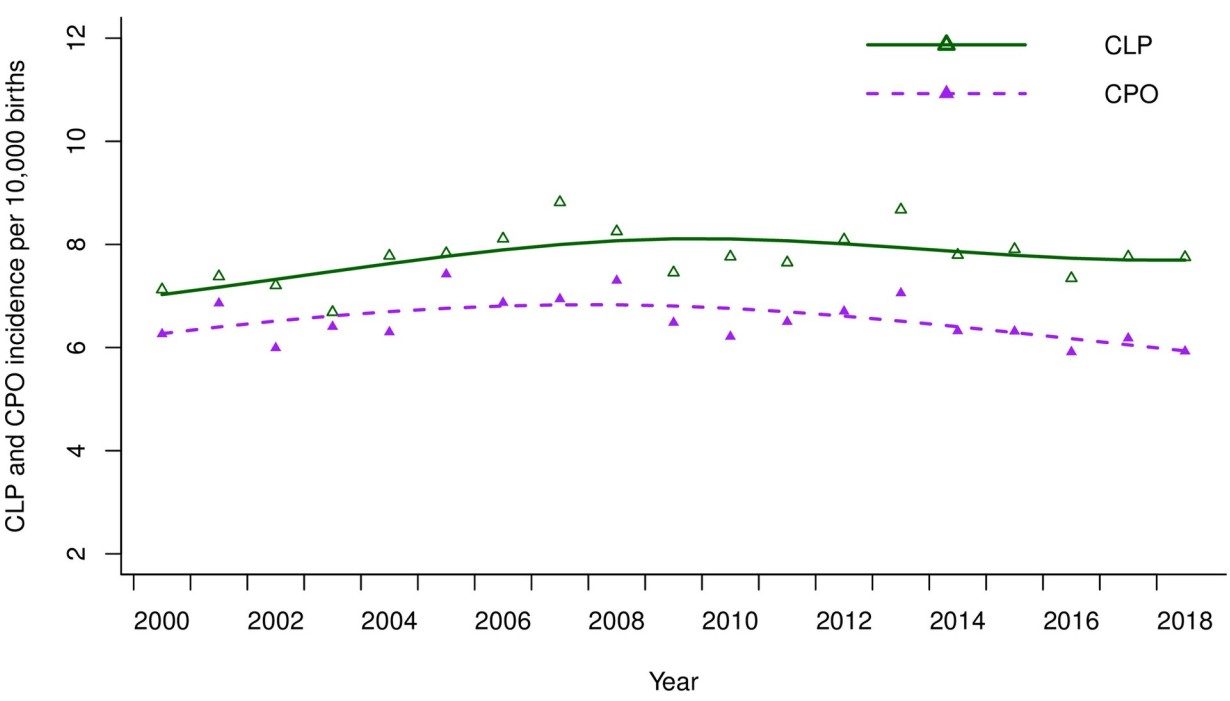

## Scotland

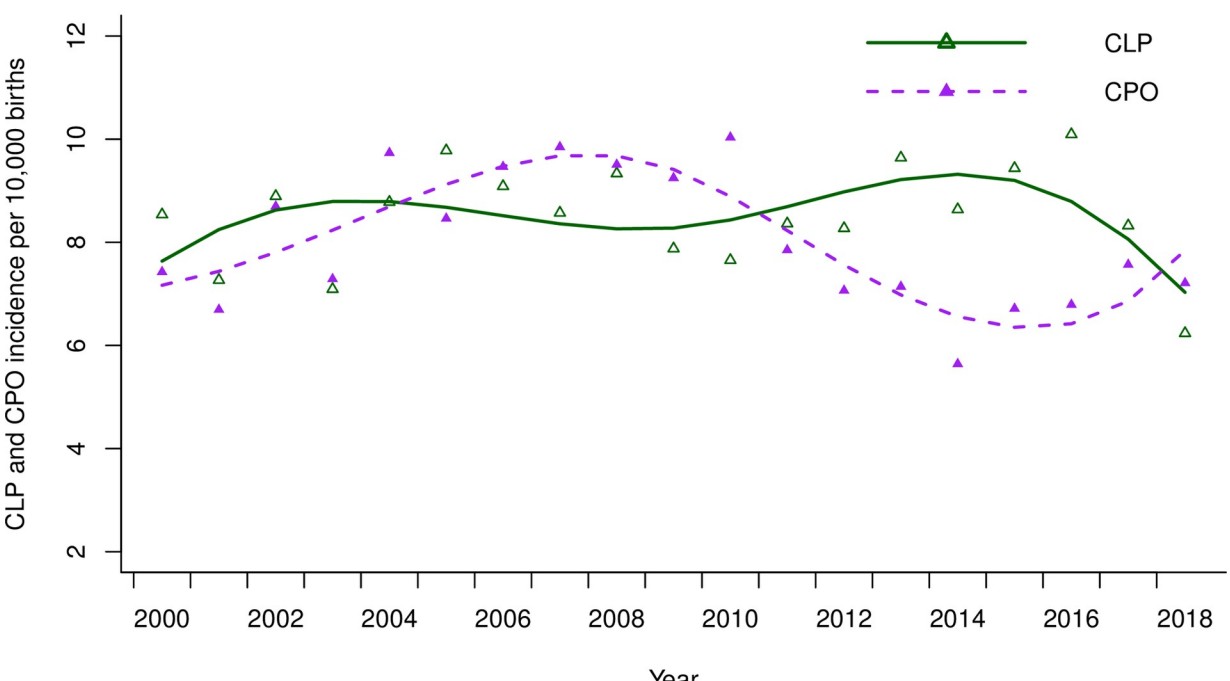

**Fig 2. Quasi-Poisson regression line of the incidence of subtypes cleft lip +- palate (CLP) and cleft palate only (CPO) over time in England Wales and Northern Ireland and in Scotland with long-term trends modelled using cubic flexible spline functions.** The scatter are the raw annual data points.

## Proxy exposure variables for proportion of active smokers in UK pregnant females

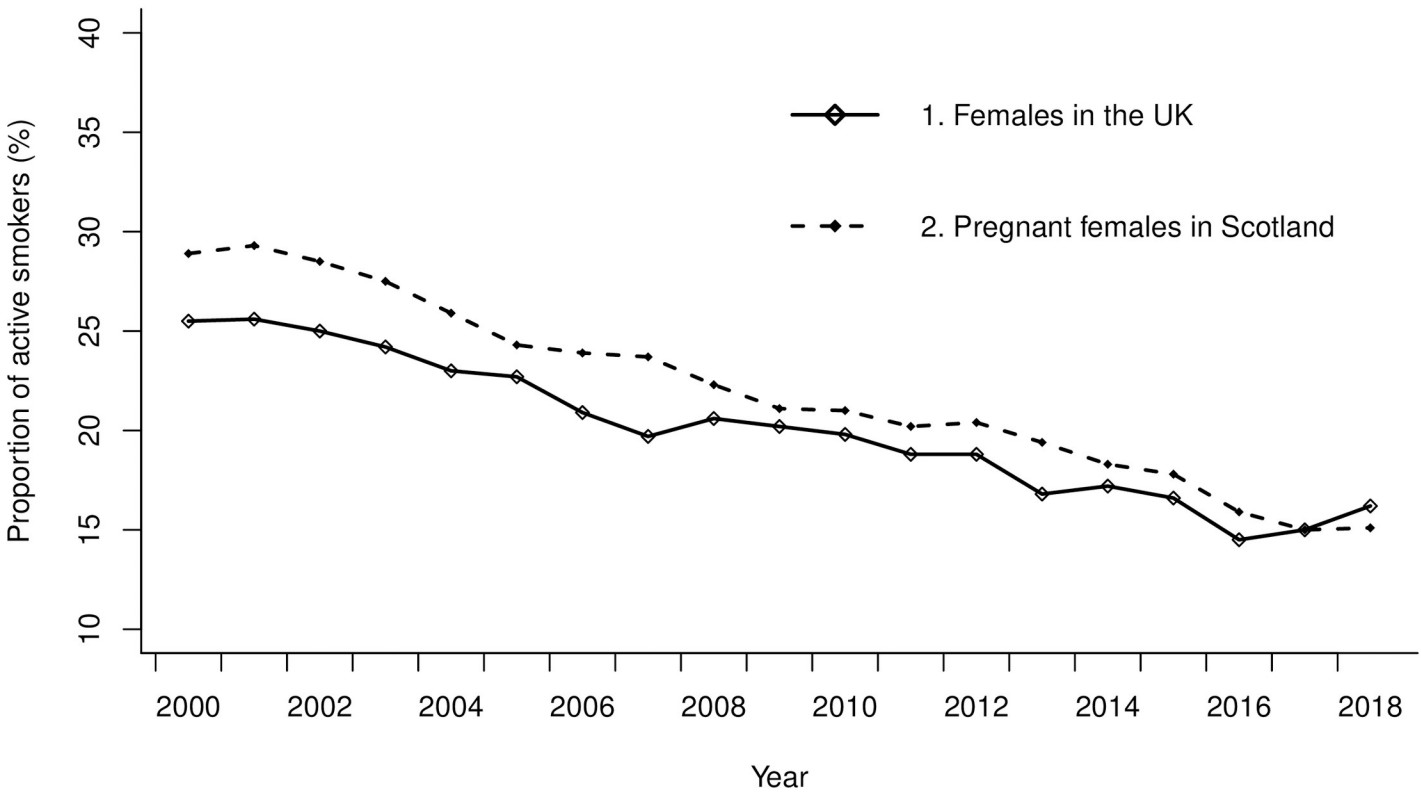

**Fig 3. Scatter plots of two proxy exposure measures for the prevalence of active smoking in pregnancy in the UK: Exposure proxy 1 is the prevalence of active smoking amongst females in the UK and Exposure Proxy 2 is the prevalence of smoking amongst pregnant women attending antenatal booking appointment in Scotland.**

following the 2007 smoke-free legislation (RR 0.92, 95%CI 0.85 to 0.99; P = 0.024) as seen in Fig 4. Sensitivity analyses to model the implementation of the smoking ban in 11 additional years in EWNI showed no evidence to suggest a decreasing level change in OFC incidence in any other year but there was a step level increase in 2003 (see S4 Appendix). Our interrupted time-series analysis strategy in EWNI scored a low risk of bias on six of the seven EPOC criteria (see Table 2). The risk to external validity from additional time varying factors occurring

**Table 1. Crude and adjusted association (Incidence rate ratio: RR) of the annual prevalence of active smoking (%), based on the two proxy measures\*, with the incidence of orofacial cleft.**

| Population of children born with Orofacial Cleft | Exposure Proxy\* | Crude RR | P Value | Adjusted RR\*\* | P Value |
|---|---|---|---|---|---|
| England, Wales and Northern Ireland | 1 | 0.997 (0.992, 1.003) | 0.380 | 0.976 (0.942, 1.012) | 0.191 |
| | 2 | 0.999 (0.994, 1.003) | 0.612 | 0.999 (0.943, 1.060) | 0.984 |
| Scotland | 1 | 1.004 (0.990, 1.018) | 0.589 | 0.989 (0.924, 1.059) | 0.759 |
| | 2 | 1.004 (0.994, 1.015) | 0.436 | 0.995 (0.877, 1.128) | 0.936 |

\*Exposure Proxy 1 is the proportion of active smokers in females over 16 years of age in the UK reported by calendar year and Exposure Proxy 2 is the proportion of active smokers in pregnant women in Scotland attending antenatal booking appointment reported by financial year.

\*\* adjustment for secular trend, maternal age and a one-year lag of exposure effect.

## England, Wales and Northern Ireland

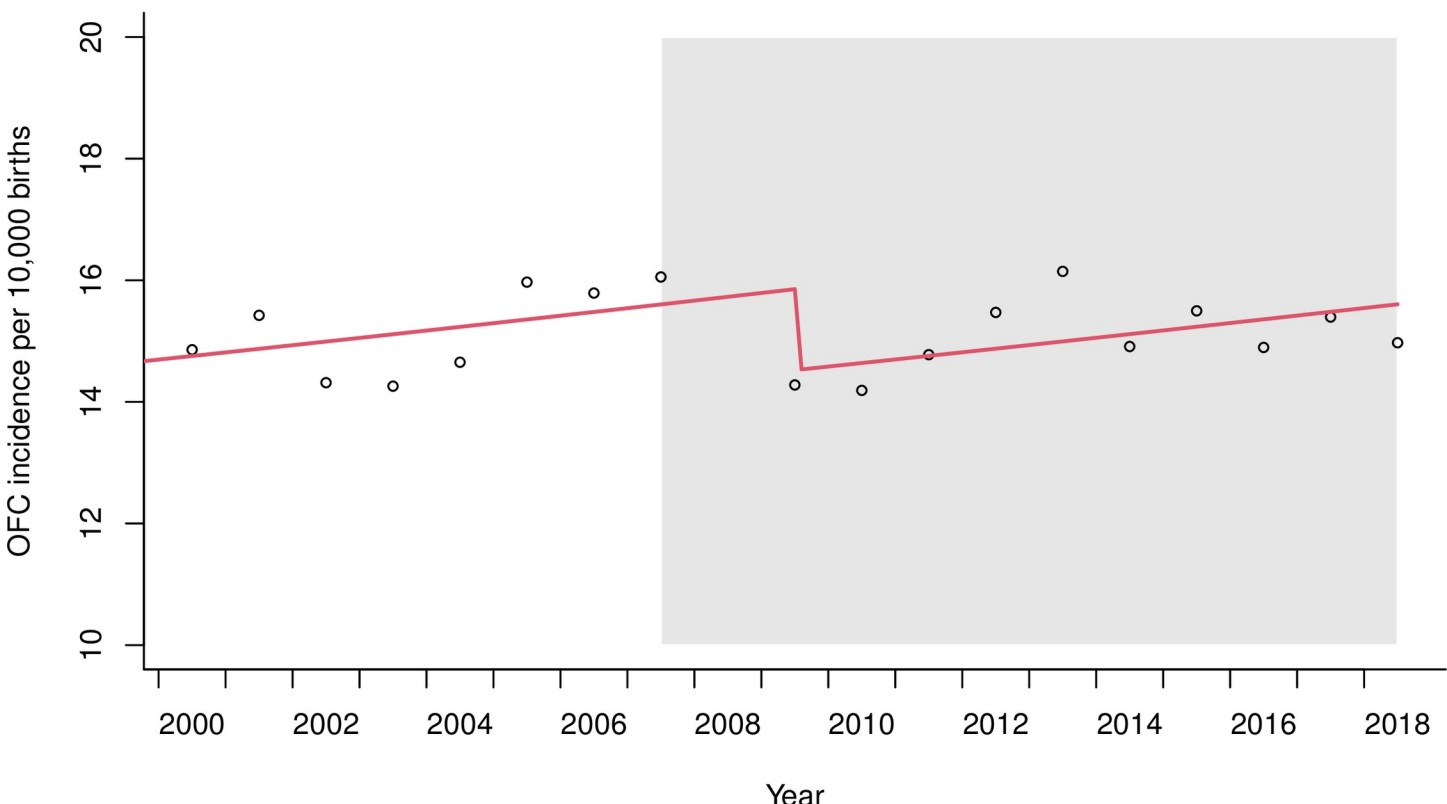

**Fig 4. Interrupted time series regression analysis of orofacial cleft (OFC) incidence in England, Wales and Northern Ireland.** Smoke-free legislation to prohibit smoking in the workplace and enclosed public spaces was implemented in 2007 and the period following is shaded in grey. Data from 2008 were omitted from the regression due to involvement in the lag phase. The first post-legislation datapoint was 2009.

during the study period, such as alcohol legislation and the global financial crash, formed the major source of bias (See S5 Appendix).

In Scotland the model showed weak evidence for change in level in the opposite direction (RR 1.16, 95%CI 0.94 to 1.44; P = 0.173) following the 2006 smoke-free legislation. Sensitivity analysis showed evidence for step changes in several years before and after legislation implementation (See S4 Appendix). The interrupted time-series analysis in Scotland had the same threats to external validity as EWNI, but there was additional risk of bias due to the change in reporting of cleft counts before and after the intervention and unknown case ascertainment (See Table 2).

### Analysis of orofacial cleft subtypes

The ecological time series analysis did not provide convincing evidence of an association between either CLP or CPO incidence and smoking prevalence in the adjusted analysis for both EWNI and Scotland (see S6 Appendix). In the natural experiment in EWNI and Scotland there was weak evidence to suggest step level changes in the incidence of both CLP and CPO and the presence of multiple additional step changes in the sensitivity analysis suggest the results were not robust (see S4 Appendix).

**Table 2. Cochrane Effective Practice and Organisation of Care (EPOC) [42] criteria to assess the risk of bias in interrupted time series studies in the datasets for England, Wales and Northern Ireland (EWNI) and for Scotland.**

| EPOC Criteria | Description in our study | Risk of bias in EWNI | Risk of bias in Scotland |
|---|---|---|---|
| 1. Intervention independent of other changes | National trends, events, guidelines and policies relating to co-variables may have influenced the outcome of orofacial cleft incidence and were a threat to external validity. These have been considered chronologically in S4 Appendix | High | High |
| 2. Shape of the intervention effect pre-specified | The anticipated impact of the intervention was predicted a priori in terms of lag phase, change in level and change in slope with rationale explained | Low | Low |
| 3. Intervention unlikely to affect data collection | The source of cleft data from the CRANE Database in EWNI was the same before and after the intervention. There was a change in the reporting of cleft counts in Scotland during the study period (before 2010 by financial year and after 2010 by calendar year) | Low | High |
| 4. Primary outcome measure measured objectively | Cleft data is entered into national registries in a systematic and objective way. Researchers in this study were blind to this process | Low | Low |
| 5. Incomplete data adequately addressed | Case ascertainment from the CRANE Database in EWNI known to be 95%. Case ascertainment in the clinical database of the National Cleft Service for Scotland not known | Low | High |
| 6. Selective outcome reporting | The study reported all outcomes (i.e. all cleft types) | Low | Low |
| 7. Appropriate analysis using interrupted time series techniques | Segmented regression technique used as previously described in the literature | Low | Low |

## Discussion

### Statement of principal findings

Smoking behaviour changed dramatically in the UK between 2000 to 2018, with the proportion of active smokers reducing by more than a third. Even with these dramatic reductions in smoking prevalence, the predicted impact on OFC incidence in the study period is relatively small, as illustrated in our PAF predictions. In our ecological time series, there was little evidence of an association between the prevalence of active smoking and the incidence of OFC in either EWNI or Scotland at a population level. The national implementation of smoke-free legislation set the scene for a natural experiment and in EWNI the interrupted time-series model suggested an 8% (1 to 15%; P = 0.024) relative step level reduction in OFC incidence following the implementation of the ban. The absence of level change reductions in other years within the sensitivity analyses in EWNI offers some reassurance that the findings reflect an underlying population change in OFC risk factors, rather than a chance statistical finding. However, the results from the natural experiment in Scotland did not replicate those seen in EWNI.

### Strengths and weaknesses of the study

The main strength of this study was the availability of national data for both outcome and exposure that had been collected in a systematic way throughout the study period. The ecological study design is advantageous because it offers the opportunity to examine the contextual effect of an environmental risk factor on a health outcome in complex real-world situations and without the selection bias associated with groups of cases in individual-level studies [49]. The natural experiment is a strong quasi-experimental research design in this setting because there was a clear intervention and the pre- and post-intervention time periods were well differentiated and similar in length [9,50–52].

Our two proxy measures of the proportion of active smokers amongst pregnant women in the UK had similar trends over time and were therefore likely to be representative of the trends

in our target population but were not able to provide information on the multidimensional nature (including intensity and duration) of smoking behaviour. The ecological fallacy has been well described and precludes individual-level inference being applied [19]. For the natural experiment, our two main assumptions were the linearity of the pre-ban trend and the stability of population characteristics and exposures over time [43]. In addition, we assumed that the smoke-free legislation would reduce exposure to passive smoking and whilst this has been demonstrated in public places, there has been a suggestion that smoke-free legislation may lead to an increase in passive smoking in the private or home environment [53]. We attributed observed changes to the intervention, but implementation of additional legislation in the devolved UK nations during the study period (such as increasing the legal age to buy tobacco in England in 2007 [54] and the regulation of alcohol sales in Scotland in 2010 [55]) present a threat to external validity [56]. In addition, the socioeconomic impact of the global financial crash in 2008 may have influenced the distribution of births and ultimately OFC aetiology [57] although the evidence for any impact on perinatal outcomes following the crash is weak [58].

Power to detect an effect if one exists in a time-series study is dependent on the number of data points, stability of the data points, strength of effect, autocorrelation and overdispersion [8,47,59]. The number of data points in our study (limited by the availability of OFC data), small anticipated effect size and presence of overdispersion was insufficient to estimate the ecological correlation between smoking and OFC incidence with an informative level of precision [43,51,60].

## Comparison with previous studies

The incidence of OFC has been described in epidemiological studies for many populations [61–67]. Time-series analysis techniques have been used to evaluate the change in OFC incidence in Slovenia over a 26-year period, with unexplained observation of cycling and clustering of cases [68]. Examples of ecological time-series to evaluate the association of environmental exposures and health outcomes have most commonly been performed in the field of air pollution and mortality, where daily data is publicly available [69,70]. Two previous ecological studies using data from Denmark and USA did not find an association between reducing prevalence of active smoking and OFC incidence [71,72].

Natural experiment methodology has been used frequently in econometric evaluations, but the translation to health epidemiology is difficult because the interventions being studied are often more subtle [9,73,74]. The Dutch Hunger Winter served as an important example of a natural experiment for birth outcomes, where maternal pre-natal nutrition could be linked to offspring schizophrenia [75]. The impact of smoke-free legislation has shown level change reductions in non-cleft areas of healthcare outcomes, with most evidence for acute coronary events, where there is a pooled level change reduction of 10% [16,76–79]. The impact of smoke-free legislation on prenatal outcomes is inconsistent [16], but in England has been associated with a 4% reduction in low birth weight [41], in Scotland and the Netherlands a 4–5% reduction in small for gestational age [80,81] and a pooled reduction of 10% in pre-term births across three continents [82]. We are not aware of a previous study to investigate the impact of smoke-free legislation on OFC incidence.

## Interpretation and implications

The findings from this study can be considered in the context of existing knowledge on cleft aetiology and triangulated with results from studies with alternative methodologies. Cigarette smoking is one of many modifiable environmental risk factors that is thought to influence cleft aetiology [2,30]. The mechanism of action of smoking may take effect either through direct

interaction with neonatal tissues or through gene-environment interactions [83,84]. The pooled effect estimate from the meta-analysis of individual-level observational studies was used to project OFC incidence in this study, based on active smoking prevalence and assuming no involvement of confounding factors [4]. The striking finding was that despite a relative reduction in active smoking prevalence of 36–48% in the exposure proxies during the study period, the projected relative reduction in OFC incidence was between 2–4%. It is important to note that from a public health perspective the complete elimination of active smoking at the start of this study in EWNI, where OFC incidence was 14.8 per 10,000 births, would only have projected a decrease of 1.3 per 10,000 births over the study period and we cannot rule out a change of this magnitude in the data we have.

The interpretation of the absent association between OFC incidence and active smoking prevalence in our ecological study is consistent with data from individual studies in our meta-analysis [4] and we lacked power in both EWNI and Scottish data to be able to identify such a small association. We cannot rule out a moderate causal relationship, that may have been masked by annual variation, the influence of confounders or imprecision in the estimates over time. The data are, therefore, compatible with no effect or a moderate effect (as reported in previous observation studies) but not a larger effect.

The level change reduction in OFC incidence observed in the EWNI interrupted time-series corresponded with the implementation of smoke-free legislation and may indicate that a change in exposure to passive smoking was associated with a reduced number of children born with orofacial clefts. Intuitively, active smoking might be expected to have a larger effect on OFC aetiology than passive smoking, but meta-analysis of individual-level data does not show this to be the case [5]. Furthermore, passive smoking might affect a higher proportion of mothers in the UK and in working environments with exposure to passive smoke for a long duration, the dose could exceed that of females smoking a small number of cigarettes per day. The incline of the slope before and after the legislation and the step level increase observed in the 2003 sensitivity analysis may reflect random variations in OFC incidence or could be linked to increases in other environmental covariables during the study period such as maternal age [26,27,29] or body mass index [85,86].

A level change was not observed in the Scottish data at the time of smoke-free legislation implementation, but step changes in both directions were observed in additional years during the sensitivity analysis. This may reflect the reduced power and increased variability of the smaller Scottish sample size, different smoking behaviours in Scotland or the influence of unique policies from the devolved government on potential confounding factors. Level change reductions have been demonstrated in Scotland following smoke-free legislation for conditions with a higher frequency of occurrence than OFC, such as small for gestational age and hospital admissions for childhood asthma [78,80].

CLP and CPO subtypes share risk factors but have been shown to have distinct epigenetic profiles and have historically been studied in the literature as separate entities [87]. There have been conflicting reports from individual-level observational studies about the relative effect estimates of active maternal smoking with these cleft subtypes, but our meta-analysis suggests overlapping confidence intervals [4]. In EWNI the incidence of CLP and CPO both reduced following the implementation of the smoke-free legislation, but we were underpowered to test for any difference.

## Conclusions

These population-level time series studies make use of the systematic collection of both exposure and outcome data in the UK and facilitate a novel insight into the relationship between cigarette smoking and OFC incidence. The data are suggestive of a potentially important

public health benefit with a reduction in cleft incidence following the implementation of the 2007 smoke-free legislation in England, Wales, and Northern Ireland, but this was not replicated in data from Scotland. The findings from this study can be triangulated with previous observational studies to expand and strengthen our knowledge of the modifiable influences on OFC aetiology. Replication of this study within different populations would provide additional confidence for the inferences described [9]. In addition, alternative approaches, such as utilisation of instrumental variables in a Mendelian Randomisation design, are required to further supplement and overcome the limitation of traditional methods.

## Supporting information

**S1 Appendix. Projections of orofacial cleft incidence.**
(DOCX)

**S2 Appendix. Data set.**
(XLSX)

**S3 Appendix. R code.**
(R)

**S4 Appendix. Natural experiment sensitivity analyses.**
(DOCX)

**S5 Appendix. Threats to external validity in the natural experiment.**
(DOCX)

**S6 Appendix. Association between the incidence of orofacial cleft subtypes and active smoking prevalence.**
(DOCX)

## Acknowledgments

Lyndsay Kirk for her work amalgamating the clinical database of the National Cleft Service for Scotland.

## Author Contributions

**Conceptualization:** Matthew Fell, Jonathan Sandy, Andrew Wills, Sarah J. Lewis.

**Data curation:** Matthew Fell, Craig Russell, Jibby Medina, Toby Gillgrass, Jonathan Sandy, Sarah J. Lewis.

**Formal analysis:** Matthew Fell, Craig Russell, Toby Gillgrass, Shaheel Chummun, Alistair R. M. Cobb, Jonathan Sandy, Yvonne Wren, Andrew Wills, Sarah J. Lewis.

**Funding acquisition:** Matthew Fell, Yvonne Wren, Sarah J. Lewis.

**Investigation:** Matthew Fell, Craig Russell, Jibby Medina, Jonathan Sandy, Andrew Wills, Sarah J. Lewis.

**Methodology:** Matthew Fell, Craig Russell, Shaheel Chummun, Alistair R. M. Cobb, Jonathan Sandy, Yvonne Wren, Andrew Wills, Sarah J. Lewis.

**Project administration:** Matthew Fell, Sarah J. Lewis.

**Resources:** Matthew Fell, Andrew Wills.

**Software:** Matthew Fell.

**Supervision:** Craig Russell, Jibby Medina, Toby Gillgrass, Shaheel Chummun, Alistair R. M. Cobb, Jonathan Sandy, Yvonne Wren, Andrew Wills, Sarah J. Lewis.

**Validation:** Matthew Fell, Craig Russell, Jonathan Sandy, Andrew Wills, Sarah J. Lewis.

**Visualization:** Matthew Fell, Jonathan Sandy, Andrew Wills, Sarah J. Lewis.

**Writing – original draft:** Matthew Fell.

**Writing – review & editing:** Matthew Fell, Craig Russell, Jibby Medina, Toby Gillgrass, Shaheel Chummun, Alistair R. M. Cobb, Jonathan Sandy, Yvonne Wren, Andrew Wills, Sarah J. Lewis.

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
