## [Decision Letter · Decision Letter 0]

2 Sep 2021

PONE-D-21-23595

The impact of changing cigarette smoking habits and smoke-free legislation on orofacial cleft incidence in the United Kingdom: evidence from two time-series studies

PLOS ONE

Dear Dr. Fell,

Thank you for submitting your manuscript to PLOS ONE. After careful consideration, we feel that it has merit but does not fully meet PLOS ONE’s publication criteria as it currently stands. Therefore, we invite you to submit a revised version of the manuscript that addresses the points raised during the review process.

We look forward to receiving your revised manuscript.

Kind regards,

Sinan Kardeş, M.D.

Academic Editor

PLOS ONE

Journal Requirements:

Reviewers' comments:

Reviewer's Responses to Questions

**Comments to the Author**

1. Is the manuscript technically sound, and do the data support the conclusions?

Reviewer #1: Yes

2. Has the statistical analysis been performed appropriately and rigorously? 

Reviewer #1: Yes

3. Have the authors made all data underlying the findings in their manuscript fully available?

Reviewer #1: Yes

4. Is the manuscript presented in an intelligible fashion and written in standard English?

Reviewer #1: Yes

5. Review Comments to the Author

Reviewer #1: General comments

Dear authors, thank you for the opportunity to read your work. This paper aims to investigate the effect of active and passive smoking on the orofacial clefts incidence in the UK. To do this, the authors adopted the state-of-the-art of regression and time-series analysis. In this regard, I congratulate the authors for the extensive comparison with the existing literature, the transparency, completeness, and clarity of the methodology, and the detailed analysis of possible biases. Finally, the authors conclude that a significant 8% reduction occurred in England, Scotland, and Northern Ireland, while this did not occur in Scotland. At present, I believe there are only a few aspects to be clarified before proceeding with the publication.

Major comments

1) Section: Methods, Statistical methods.

1.1. A brief explanation of the use of the p-value should be provided. In particular, has a significance threshold been adopted? If so, specify which one. Or (recommended, https://www.ncbi.nlm.nih.gov/pmc/articles/PMC4877414/) were p-values used as graded measures of the strength of the evidence against the null hypothesis?

1.2. Similarly, a brief explanation should be provided as to the strength of the correlations (e.g., RR). In particular, were “intensity” thresholds used or, in this case too, were the results evaluated using a continuous scale?

1.3. In some cases, in the “Results” section, the authors showed IQRs and, in others, the mean values. Was this done in consideration of the distributive nature of the data (i.e., non-Gaussians and Gaussians, respectively)? I suggest specifying this detail in the “Methods” section.

2) Section: Conclusions. For completeness and transparency, I suggest also reporting the negative result obtained for Scotland (i.e., the absence of evident correlations).

Minor comments

m1) Section: Abstract, Lines 42-46. Please also provide values for negative results.

m2) Section: Methods, Lines 184-187. It is not clear why - although a decreasing trend was already present - we should not have expected a change in slope after the smoking ban implementation. In particular, why is this not supposed to have increased the reduction “speed”?

m3) Section: Figures 1, 4. I suggest specifying the quantity reported on the y-axis (e.g., Figure 1, "OFC incidence [...]"). In this way, the graphs will be clear regardless of the caption.

m4) Section: Results, Line 276. Is a "mean of 0.8%" meant?

Other comments

o1) Section: Methods, Lines 130-132. I kindly ask the authors if they can better explain what they mean by this sentence. Thank you.

o2) Section: Methods, Lines 224-226. I kindly ask why the chosen lag was exactly one year (that is, why three months more than the gestation duration? Is this simply an upward approximation since the model works with annual rates?) Thank you.

6. PLOS authors have the option to publish the peer review history of their article (what does this mean?). If published, this will include your full peer review and any attached files.

Reviewer #1: **Yes: **Alessandro Rovetta

---

## [Author Response · Author response to Decision Letter 0]

25 Sep 2021

Many thanks to Reviewer 1 for their kind comments and helpful suggestions.

Major comments

1) Section: Methods, Statistical methods.

1.1. A brief explanation of the use of the p-value should be provided. In particular, has a significance threshold been adopted? If so, specify which one. Or (recommended, https://www.ncbi.nlm.nih.gov/pmc/articles/PMC4877414/) were p-values used as graded measures of the strength of the evidence against the null hypothesis?

RESPONSE: Thank you for raising this point and for the helpful reference to Greenland et al., 2016. We resisted the use of the term ‘statistical significance’ and arbitrary dichotomous significance thresholds in the manuscript (in agreement with Greenland et al 2016 and Stern and Davey Smith 2001 https://www.ncbi.nlm.nih.gov/pmc/articles/PMC1119478). We have added an explanation of this in the statistical methods section (Page 9, lines 209-212) and hope this adds clarity. 

“Effect estimates were reported as incidence rate ratios (RR) with 95% confidence intervals (CI) and p-values. P values were interpreted as continuous measures of the strength of evidence against the null hypothesis.”

1.2. Similarly, a brief explanation should be provided as to the strength of the correlations (e.g., RR). In particular, were “intensity” thresholds used or, in this case too, were the results evaluated using a continuous scale?

RESPONSE: Thank you too for this extension to the previous point. We did not use any intensity thresholds for the incidence rate ratios for the same reason of not wanting to arbitrarily dichotomise the results. Results were evaluated using a continuous scale. We hope that the added text (as per the point above) in the statistical methods (Page 9, lines 209-212) section will sufficiently address this point.

1.3. In some cases, in the “Results” section, the authors showed IQRs and, in others, the mean values. Was this done in consideration of the distributive nature of the data (i.e., non-Gaussians and Gaussians, respectively)? I suggest specifying this detail in the “Methods” section.

RESPONSE: Thank you for highlighting this confusing mix of reporting. To improve consistency, we have discarded the means and have instead reported all descriptive statistics with an interquartile range. This has been specified in the statistical methods section (page 9, lines 202-203). 

“All variables were described initially through visualisation and summary statistics with interquartile range (IQR).”

2) Section: Conclusions. For completeness and transparency, I suggest also reporting the negative result obtained for Scotland (i.e., the absence of evident correlations).

RESPONSE: We have added a comment to this effect in the conclusion (page 22, lines 510-511).

“but this was not replicated in data from Scotland”.

Minor comments

m1) Section: Abstract, Lines 42-46. Please also provide values for negative results.

RESPONSE: this has been added (page 3, line 50)

“(RR 1.16, 95%CI 0.94 to 1.44; P=0.173)”.

m2) Section: Methods, Lines 184-187. It is not clear why - although a decreasing trend was already present - we should not have expected a change in slope after the smoking ban implementation. In particular, why is this not supposed to have increased the reduction “speed”?

RESPONSE: We did not anticipate a change in slope (or ‘speed) because the smoking ban represents a singular event leading to a one-off change in passive smoking exposure. We know from data reported in this paper (i.e. active smoking prevalence and maternal age) that other factors were changing consistently during the study period, in comparison with the sudden change in passive smoking exposure following the ban. Therefore, we anticipated a change in level after the ban but not a change in slope. The anticipated model is of course, a hypothesis, but the current literature on interrupted time series specifies this impact model as an important step that should be reported. We hope that the current succinct description of this in the text is adequate.

m3) Section: Figures 1, 4. I suggest specifying the quantity reported on the y-axis (e.g., Figure 1, "OFC incidence [...]"). In this way, the graphs will be clear regardless of the caption.

RESPONSE: The y-axis on the graphs have been amended as per suggestion (Figure 1, 2 and 4). 

m4) Section: Results, Line 276. Is a "mean of 0.8%" meant?

RESPONSE: Thank you. The means have now been discarded as per major comment and response above

Other comments

o1) Section: Methods, Lines 130-132. I kindly ask the authors if they can better explain what they mean by this sentence. Thank you.

RESPONSE: The sentence has been amended. Addition of ‘due to the change in annual reporting intervals’ to improve clarity (page 6, lines 131-132).

“Children born in Scotland between January-April 2010 were counted twice due to the change in annual reporting intervals, but exclusion of 2010 data in the analysis did not impact the results.”

o2) Section: Methods, Lines 224-226. I kindly ask why the chosen lag was exactly one year (that is, why three months more than the gestation duration? Is this simply an upward approximation since the model works with annual rates?) Thank you.

RESPONSE: Yes this is correct and has been explained in the text to improve clarity (page 10, lines 230-231)

“The final model was adjusted for the yearly long-term incidence trend and 

maternal age with a 1-year lag, as an upward approximation to account for the 9-month human gestation period since the model works with annual rates.”

---

## [Decision Letter · Decision Letter 1]

27 Oct 2021

The impact of changing cigarette smoking habits and smoke-free legislation on orofacial cleft incidence in the United Kingdom: evidence from two time-series studies

PONE-D-21-23595R1

Dear Dr. Fell,

We’re pleased to inform you that your manuscript has been judged scientifically suitable for publication and will be formally accepted for publication once it meets all outstanding technical requirements.

Kind regards,

Sinan Kardeş, M.D.

Academic Editor

PLOS ONE

Additional Editor Comments (optional):

Reviewers' comments:

Reviewer's Responses to Questions

**Comments to the Author**

1. If the authors have adequately addressed your comments raised in a previous round of review and you feel that this manuscript is now acceptable for publication, you may indicate that here to bypass the “Comments to the Author” section, enter your conflict of interest statement in the “Confidential to Editor” section, and submit your "Accept" recommendation.

Reviewer #1: All comments have been addressed

2. Is the manuscript technically sound, and do the data support the conclusions?

Reviewer #1: Yes

3. Has the statistical analysis been performed appropriately and rigorously? 

Reviewer #1: Yes

4. Have the authors made all data underlying the findings in their manuscript fully available?

Reviewer #1: Yes

5. Is the manuscript presented in an intelligible fashion and written in standard English?

Reviewer #1: Yes

6. Review Comments to the Author

Reviewer #1: Dear Authors,

Thank you for your professional revisions. All the points I raised have been adequately addressed. Therefore, I'm glad to vote for the publication of the manuscript. I wish you the best for this and your future research.

7. PLOS authors have the option to publish the peer review history of their article (what does this mean?). If published, this will include your full peer review and any attached files.

Reviewer #1: **Yes: **Alessandro Rovetta

---

## [Editor Report · Acceptance letter]

29 Oct 2021

PONE-D-21-23595R1 

The impact of changing cigarette smoking habits and smoke-free legislation on orofacial cleft incidence in the United Kingdom: evidence from two time-series studies 

Dear Dr. Fell:

I'm pleased to inform you that your manuscript has been deemed suitable for publication in PLOS ONE. Congratulations! Your manuscript is now with our production department. 

Kind regards, 

on behalf of

Dr. Sinan Kardeş 

Academic Editor

PLOS ONE